# An 8q24 Gain in Pancreatic Juice Is a Candidate Biomarker for the Detection of Pancreatic Cancer

**DOI:** 10.3390/ijms24065097

**Published:** 2023-03-07

**Authors:** Iris J. M. Levink, Malgorzata I. Srebniak, Walter G. De Valk, Monique M. van Veghel-Plandsoen, Anja Wagner, Djuna L. Cahen, Gwenny M. Fuhler, Marco J. Bruno

**Affiliations:** 1Department of Gastroenterology & Hepatology, Erasmus MC, University Medical Center, 3015 GD Rotterdam, The Netherlands; 2Department of Clinical Genetics, Erasmus MC, University Medical Center, 3015 CN Rotterdam, The Netherlands

**Keywords:** cfDNA, copy number variant, CNV, liquid biopsy, NIPT

## Abstract

Secretin-stimulated pancreatic juice (PJ), collected from the duodenum, presents a valuable biomarker source for the (earlier) detection of pancreatic cancer (PC). Here, we evaluate the feasibility and performance of shallow sequencing to detect copy number variations (CNVs) in cell-free DNA (cfDNA) from PJ for PC detection. First, we confirmed the feasibility of shallow sequencing in PJ (n = 4), matched plasma (n = 3) and tissue samples (n = 4, microarray). Subsequently, shallow sequencing was performed on cfDNA from PJ of 26 cases (25 sporadic PC, 1 high-grade dysplasia) and 19 controls with a hereditary or familial increased risk of PC. 40 of the 45 PJ samples met the quality criteria for cfDNA analysis. Nine individuals had an 8q24 gain (oncogene MYC; 23%; eight cases (33%) and one control (6%), *p* = 0.04); six had both a 2q gain (STAT1) and 5p loss (CDH10; 15%; four cases (7%) and two controls (13%), *p* = 0.72). The presence of an 8q24 gain differentiated the cases and controls, with a sensitivity of 33% (95% CI 16–55%) and specificity of 94% (95% CI 70–100%). The presence of either an 8q24 or 2q gain with a 5p loss was related to a sensitivity of 50% (95% CI 29–71%) and specificity of 81% (95% CI 54–96%). Shallow sequencing of PJ is feasible. The presence of an 8q24 gain in PJ shows promise as a biomarker for the detection of PC. Further research is required with a larger sample size and consecutively collected samples in high-risk individuals prior to implementation in a surveillance cohort.

## 1. Introduction

Pancreatic cancer (PC) has a poor prognosis, with incidence rates closely mirroring mortality rates. Surveillance of individuals at risk of developing PC aims at detecting disease at an earlier (i.e., resectable) stage, but this has been proven challenging based on imaging alone [1]. As the development of PC from the first genetic alteration to cancer is expected to take years, biomarkers may enable the detection of resectable cancer stages, or preferably premalignant lesions, not yet visible on imaging. 

The presence of chromosomal aberrations is a common molecular feature in human cancer, including loss of tumor suppressor genes or gain of oncogenes, driving oncogenic signaling and cancer development. Specifically, in PC, the amplification of genes involved in DNA repair and tyrosine kinase signaling are associated with poor survival [2]. The detection of such alterations in cell-free DNA (cfDNA) released from tumor cells (circulating tumor DNA (ctDNA)) has shown promise for several cancers in (pre-) clinical studies [3,4].

Clinical testing for chromosomal aberrations in cfDNA is currently routinely performed during pregnancy by noninvasive prenatal testing (NIPT). One approach is based on shallow whole-genome sequencing of cfDNA present in maternal plasma, which consists of both maternal and fetal DNA. While an NIPT test aims to diagnose chromosomal aberrations in fetal DNA, different chromosomal patterns in maternal DNA have been incidentally detected, which raised the suspicion of maternal malignancy [5,6]. The majority of these cancers were of hematologic origin, yet solid tumors, such as colorectal, gastric, breast and ovarian cancer, were also detected [5,6,7]. The recently published TRIDENT-2 study showed a prevalence of 70% maternal malignancy in individuals with two or more CNVs [8]. Remarkably, these aberrations originated from undiagnosed maternal cancer, suggesting that this test may serve as an early screening tool for cancer.

For the detection of PC, pancreatic juice (PJ) may provide a promising biomarker source, as it is in close contact with pancreatic ductal cells, which are the cells of origin for more than 90% of PCs [9,10]. PJ harbors higher concentrations of cfDNA than plasma [11] and, in contrast to fine-needle biopsy (FNB), it is expected to contain information from all tumor clones present in the pancreatic ductal system. A wash-out of PJ from the pancreatic ductal system can be provoked by secretin and collected by suction through an endoscope from within duodenum, which is relatively noninvasive and carries a very low procedural risk compared to tissue sampling by endoscopic ultrasound (EUS)-guided FNB or PJ collection through direct pancreatic duct cannulation.

The aim of this study was to evaluate the feasibility of the shallow sequencing of cfDNA obtained from PJ and plasma from patients with PC using the clinically available and robust NIPT pipeline. Subsequently, we compared the presence of chromosomal aberrations in the PJ between the PC cases and controls undergoing surveillance for a hereditary predisposition of PC.

## 2. Results

### 2.1. Feasibility Phase

To investigate the feasibility of detecting chromosomal instability in PJ, a pilot study was performed including four patients with HGD (n = 1) or PC (n = 3). The concentration of the three PC plasma samples ranged from 0.05 to 0.11 ng/µL and of the four PJ samples from 11 to 33 ng/µL. After multiple concentration (for plasma) and dilution (for PJ) steps, the input cfDNA concentration of the three plasma samples ranged from 52 to 74 ng/µL and that of the four PJ samples from 19 to 398 pg/µL (three-quarters had a concentration of 185 pg/µL or more). cfDNA from plasma samples harbored mostly fragments of 167 bp (mode size) with the expected sawtooth pattern of 11 bp related to helical periodicity of DNA [12,13], whereas fragments in PJ did not show this typical pattern and were generally of shorter length (peak prevalence at 68–102; Figure 1A).

The number of reads in the PJ ranged from 10,000,000 to 116,000,000 (1.8–3.6% duplicates), whereas those in plasma ranged from 14,630,000 to 17,500,000 (2.5–3.8% duplicates). The fraction percentage of tumor cfDNA was 0.7–1.9% for plasma, which was too low to call chromosomal aberrations in cfDNA. The %cfDNA in the PJ samples ranged from 6–8%.

Two (PC #3, HGD #4) out of four samples from the PJ showed an amplification on the q-arm of chromosome 8 (both chr8: 128,000,000–140,000,000; 8q24), with PC #3 also showing a deletion on 10p (chr10: 10,000,000–21,000,000; 10p14–10p21.31; Figure 1B). To exclude the technical errors generated by the usage of different biomaterial (e.g., DNA fragmentation), these findings were compared to microarray results in tumor tissue (Appendix A for H&E stainings). The 8q24 gain (chr8: 128,491,792–130,491,752) of HGD (#P4) was also the only chromosomal aberration detected in tissue of this patient. Tissue of PC #3 showed a very noisy profile without significant chromosomal aberrations. However, for this patient, the time between PJ collection and pancreatic resection had been four months, during which the patient had undergone neoadjuvant chemoradiotherapy. A partial response to treatment was seen as a severely fibrotic area of 3 cm (ypT2N1), which may have hampered tumor DNA extraction from this tissue.

Based on these results, we decided to evaluate the feasibility of the shallow sequencing of PJ in a larger group.

### 2.2. Experimental Phase

PJ samples from 45 individuals: 26 Cases (25 PC, 1 HGD) and 19 controls were included in the experimental phase of the study (Table 1). The cases were of older age (69 (IQR 9) vs. 60 (IQR 9) years, *p* < 0.001) and more often males (n = 18 (69%) vs. n = 7 (37%), *p* = 0.04). The majority of the cases was treatment-naïve and underwent PJ collection at the time of diagnosis (n = 23, 88%), while three had undergone their last chemotherapy 2–4 weeks before collection (#PC08, #PC10 and #PC15) and underwent PJ collection during an EUS procedure that was indicated for the fiducial placement prior to radiotherapy (and surgery). Sixteen cases (62%) had locally advanced disease, fifteen (58%) had a solid mass located in the pancreatic head and seven (27%) had a common bile duct stent in situ at the time of collection. After 35 months (IQR 11) of follow-up, 11 cases (42%) had undergone surgery, and 20 (77%) had died of PC.

Of nineteen controls, twelve had a proven germline mutation: eight (18%) *CDKN2A* (p16), three (7%) *BRCA1*/*2* and one (2%) *PALB2*. The remaining seven controls were deemed at risk because they had multiple family members with PC without proven gene mutations (as previously investigated by germline genetic testing; see Appendix A for the in- and exclusion criteria). None of the controls had symptoms or a history of diabetes mellitus. Seven out of eight individuals with a CDKN2A germline mutation previously underwent curative treatment for a melanoma (<T1). One individual (#Co07) developed breast cancer two years before PJ collection, seemed in remission during collection but died of metastases (pathology-proven breast cancer) one year after inclusion in the study. Ove 25 months (IQR 12) of follow-up, none of the controls developed cancer, morphologic (pancreatic) abnormalities on imaging or worrisome symptoms, such as new-onset diabetes, jaundice or acute pancreatitis.

### 2.3. Copy Number Variants in Pancreatic Juice

For the CNV analysis of the PJ samples, the median cfDNA input concentration was 161 pg/µL (IQR 276), and 7.6 million (IQR 9.1) read pairs were generated. These values were not different between the cases and controls (*p* > 0.05). The controls more often had a fragment length (also known as insert size) of 133–135 bp compared to the cases (*p* = 0.05; Figure 2A).

After the exclusion of five individuals who did not meet the >5 million read pairs criterion, the CNV results of the remaining 40 individuals (24 cases, 16 controls) were further evaluated. Twenty samples (50%; 9/24 cases (38%), 11/16 controls (69%); *p* = 0.05) had no significant CNVs. Of these, 50% showed a clear sawtooth pattern on the insert size graph (Figure 2B) with a peak in the fragment size prevalence between 68 and 81 bp. The absence of a sawtooth pattern did not differentiate between the cases and controls (*p* < 0.05). The samples of five individuals (13%; 3/24 cases (13%), 2/16 controls (13%)) showed complex chaotic aberrations spread over multiple chromosomes without a clear relationship to each other. In addition to this complex chaotic profile, the DNA of these samples was severely fragmented and did not show a sawtooth pattern (Figure 2C). Fifteen individuals (12/24 cases, 3/16 controls) showed clues for chromosomal aberrations that could be subdivided into either an 8q24 subgroup of samples showing at least an 8q24 gain (with or without a 10p loss and/or 18q loss) or a subgroup showing at least a 2q gain combined with a 5p loss. The additional aberrations present in this latter subgroup were a 7q gain, 12q gain and/or 16q loss. The chromosomal aberrations in 8q24 and 2q-5p were mutually exclusive.

#### 2.3.1. The 8q24 Subgroup

Nine individuals (23%; eight cases (33%), one control (6%), *p* = 0.04) had at least an 8q24 gain (chr8: 128,000,000–146,000,000; Figure 3A,B). The DNA insert size mode was 100–120bp in 7/9 samples (69 bp and 170 bp in the other two), and all graphs showed a smooth shape (Figure 2D). Other CNVs present in this subgroup were a 10p loss (n = 5; chr10: 8,000,000–25,000,000; 10p11–1p15; Figure 3A,C), 18q loss (n = 2; chr18: 45,000,000–67,000,000; 18q21–18q22; Figure 3A,D) and a single 5q gain (chr5: 143,000,000–155,000,000; 5q31–5q33). The presence of an 8q24 gain differentiated the cases from the controls with a sensitivity of 33% (95% CI 16–55%), specificity of 94% (95% CI 70–100%) and accuracy of 57% (95% CI 41–73%; Figure 3A and Table 2).

While the genetic variation of the 8q24.21 band has been associated with various cancer types, protein-coding genes within this band are sparse (Appendix A). *MYC,* a well-described oncogene, received our specific interest, as it is overexpressed in 44% of PC tissues [14]. *MYC* is downstream of the RAS/RAF pathway but also other oncogenic pathways (WNT-*β-*Catenin, JAK/STAT, TGF-β and Notch). *MYC* expression, in turn, drives cell growth and proliferation by binding to the enhancer box of transcription factors and acting as transcription factor to oncogenes such as *BCL2* (loss of this gene was seen in two individuals from the 8q24 subgroup), *TP53* and *p19ARF*. Tumorigenic effects of the 8q24 gain in PC may also be caused by neighboring oncogenic sequences coding for long noncoding RNAs (lncRNAs; PCAT1, CASC19, PRNCR1, CCAT1, CASC8, CCAT2, CASC11, PVT1, TMEM75 and CCDC26, all associated with various cancers) or microRNAs (miRNAs/miR; miR-1204, miR-1205, miR-1206, miR-1207-5p, miR-1207-3p and miR-1208). For instance, miR1208 was shown to be overexpressed in PJ from PC patients compared to controls [15]. Another candidate gene (located next to *MYC*) is *PVT1*, a transcriptional activator of *MYC.* Another amplified region on chromosome 8 in the PJ samples (yet not in the #P4 tissue sample) is the band 8q24.3, the location of multiple protein-coding genes. Examples are *PARP10*, *PSCA*, *HSF1* and *PLEC1*, the protein levels of which are shown to be overexpressed in the PC tissue (compared to healthy tissue) and promote tumorigenesis (by different pathways). Currently, the efficacy of therapy against proteins, which these genes code for, is being investigated for PC and multiple other cancer types [16,17,18,19,20,21].

Our data also showed a 10p loss (10p15.1–10p11.22; Appendix A) in these PJ samples. This region contains multiple genes that have been related to carcinogenesis, while one gene located on 10p12.2 may be of particular interest: transcription factor *PTF1A*. The silencing of *PTF1A* protects acinar cells during injury, which allows them to recover [22]. However, in the case of oncogenic insults, the loss of *PTF1A* potentiates acinar-to-ductal metaplasia and development of PanIN [23,24,25]. *BMI1*, a candidate gene located close to *PTF1A*, is a key player in the regulation of pancreatic β- and acinar cell proliferation. The gene is required for regeneration (e.g., after pancreatitis) [26,27,28], and inhibition of *BMI1* has been shown to upregulate the production of reactive oxygen species [29], which is an essential step for the onset of pancreatic carcinogenesis [30].

Two samples showed an 18q21 loss. This band, which is frequently altered in gastrointestinal cancers [31], harbors *SMAD4* (Appendix A). *SMAD4* expression is decreased in 58% of the PC cases, and the loss of *SMAD4* via homozygous deletion or mutation often occurs in late-stage PC [32,33]. The loss of *SMAD4* promotes carcinogenesis by the stimulation of TGF-β signaling. *BCL2*, also located on 18q21, is an anti-apoptotic protein under the modulation of *TP53,* a gene that is mutated in 71% of patients with PC [34,35]. Its expression is found to be downregulated in PC tissue, and the loss of *BCL2* has been associated to a poor survival [36,37]. A third gene of interest in this region is *DCC*, a gene that codes for the netrin-1 receptor, which has been elaboratively investigated as a tumor suppressor gene for colorectal cancer and has been associated with tumor stage in PC [38].

#### 2.3.2. The 2q-5p Subgroup

Six individuals (15%; four cases (17%), two controls (13%); *p* = 0.72) had a 2q gain (with two different areas of amplification; chr2: 150,000,000–168,000,000 and 183,000,000–195,000,000; 2q23–2q24, 2q32; Figure 3A,E) and 5p loss (chr5: 19,000,000–33,000,000; 5p13–5p14; Figure 3A,F). The DNA fragments had a mode size of 170–180 bp; this peak prevalence was preceded by a plateau that showed a notable sawtooth pattern (Figure 2E). Of these, two samples showed an additional 7q gain (chr7: 75,000,000–90,000,000; 7q11–7q21; Figure 3A,G), three a 12q gain (ca. chr12: 83,000,000–95,000,000; 12q21–12q23; Figure 3A,H) and three a 16q loss (chr16: 55,000,000–68,000,000; 16q12–16q22; Figure 3A,I). The ability to differentiate the cases from the controls was lower than for the 8q24 group (sensitivity: 17% (95% CI 5–38%), specificity: 87% (95% CI 62–98%) and accuracy: 45% (95% CI 29–62%); Table 2). The presence of either an 8q24 or 2q-5p profile had a sensitivity of 50% (95% CI 29–71%), specificity of 81% (54–96%) and accuracy of 63% (95% CI 46–77%).

Six individuals had a 2q gain, located on 2q23–2q24 or 2q32 (Figure 3A; Appendix A). The 2q24 band houses two genes, *FAP* and *GALNT3*, involved in the desmoplastic and immunosuppressive microenvironment, respectively, two major hurdles to be crossed in PC research. *FAP* has been shown to be expressed in PC cells and fibroblasts and plays a pivotal role in PC desmoplasia [39,40,41]. *GALNT3* encodes for an enzyme involved with O-linked glycosylation. It is shown to be overexpressed in well-differentiated PC tissue, yet downregulated in poorly differentiated PC, and may be a marker for prognosis. *STAT1* and *STAT4* are genes located on the 2q32 band, which encode for important components of the JAK-STAT signaling pathway, which is (among other processes) involved in apoptosis and oncogenesis, having both tumor suppressive and tumor promoting functions [42]. For PC, an increased expression of *STAT1* has been related to a favorable prognosis [43,44].

Lastly, examples of potentially important genes that have been related to carcinogenesis and are located on the other aberrant bands (5p24, 7q11–7q21,12q21–12q23 and 16q12–16q22; Appendix A) are *CDH6*, *CDH9*, *CDH10*, *CDH12, CDH18* (located on 5p24, which are involved in cell differentiation, and the loss of the heterozygosity of *CDH10* was present in 24% of cases with PC [45]), *HGF* (7q21; mediates the interaction between cancer cells and pancreatic stellate cells) [46,47], *DUSP6* (12q21; a tumor suppressor and key player in the RAS/ERK signaling pathway) [48] and *HSF4* (16q22; a transcriptional factor critical for the activation of NF-κB signaling) [49].

## 3. Discussion

This study shows that shallow sequencing using the robust NIPT pipeline is feasible for cfDNA isolated from PJ collected from the duodenal lumen after secretin stimulation but not for plasma. The most notable finding was the high prevalence of an 8q24 gain in the PC group. This was even seen in a patient with HGD, suggesting that this is a relatively early aberration that may hold promise for the early detection of PC. Interestingly, individuals with an 8q24 also tended to have a distinct “smooth” fragment length profile with rather short DNA fragments. This CNV was seen in combination with the loss of 10p11–10p15 and/or 18q21–18q22. The detection of an 8q24 gain was highly specific for PC (94%) yet had low sensitivity (33%). Another prevalent finding was the combination of a gain of 2q and a loss of 5p in patients with a distinct DNA fragmentation pattern (longer fragments with a sawtooth pattern). The presence of either an 8q24 or 2q-5p profile had a sensitivity of 50%, specificity of 81% and accuracy of 63%.

The aim of this study was to find biomarkers that enable the earlier detection of PC in individuals who undergo surveillance. Moreover, the overall presence of CNVs and those on distinct locations may guide risk stratification and the frequency of surveillance. Secretin-stimulated PJ collection is, as compared to fine-needle biopsy or PJ-collection by cannulation of the pancreatic duct during endoscopic retrograde pancreatography (ERP), less invasive. In our opinion, the safety profile potentiates repetitive collections (e.g., yearly) and allows for the monitoring of (early chromosomal) changes over time that are indicative for malignant transformation. Additionally, these changes in PJ may predict the response to therapeutic agents [50,51]. For instance, PARP inhibitors and platinum agents have shown to be effective in solid tumors bearing an unstable genome (including PC) [52,53]. Another example is *BCL2* downregulation, which is associated with the restoration of sensitivity to gemcitabine [54].

The two identified patient subgroups with CNVs showed distinct features on the fragment size graphs. The 8q24 graphs were smooth with a mode size of 100–120 bp, whereas the 2q-5p subgroup had a saw tooth pattern with a mode size of 170–180 bp. The graph of the latter also showed a distinct plateau prior to the peak. cfDNA is generally produced by apoptosis and has a modal size of ±167 bp, which corresponds to 147 bp of the DNA wrapped around a nucleosome plus the stretch of DNA on Histone H1 that links two nucleosome cores. The shorter fragments in these samples may be due to the fact of cleavage by enzymes in the PJ. However, this does not explain the differences among the subgroups. This pattern is not expected to be a result of necroptosis or cellular secretion, as DNA fragments generated in these processes are generally larger (up to >10,000 bp for necroptosis and 1000–3000 bp for secretion) [53]. Different experiments have demonstrated smaller fragment sizes (90–150 bp) for tumor-derived cfDNA than wild-type cfDNA [55,56], which may clarify the fragment size distributions in our samples. Additionally, mitochondrial DNA (mtDNA) has shown to be more fragmented (30–100 bp) [57,58].

Previous studies investigating chromosomal aberrations in tissue showed an 8q24 gain in 24–45% of patients with PC and 27% of those with HGD [59,60,61]. Therefore, the presence of this CNV may serve as a marker to detect HGD or early PC. The combination of an 8q gain and loss of 10p and 18q has been shown before in tissue; however, the loss of 18q and 10p has also been seen without an 8q gain [59,61]. We are the first to describe chromosomal aberrations in PJ that was collected noninvasively from the duodenum after stimulation by secretin. Mateos et al. (2019) [62] performed whole-exome sequencing on PJ samples collected during ERP or endoscopic nasopancreatic drainage of 39 patients with IPMN. They found 11 significantly amplified regions and 4 deleted regions. Of these, a gain of 7q21 (1/8 low-grade dysplasia (LGD), 1/20 HGD and 7/11 PC), 8q24 (2/8 LGD, 1/20 HGD and 6/11 PC) and 12q21 (2/8 LGD, 3/20 HGD and 1/11 PC) match our results.

We acknowledge several limitations to our study. The tissue samples were available only for patients included in the feasibility phase. For the three PC cases in this phase, there was ample time between PJ collection and surgery (4–10 months), during which two underwent chemotherapy (eight cycles of FOLFIRINOX). For the patient with HGD (and an 8q24 gain), only two months passed between PJ collection and surgery. Therefore, we do not have confirmation of the breakpoints on the chromosomes. However, we were able to link our results to gene expression results currently present in the literature. The low sensitivity (33%) may implicate that an 8q24 gain represents a subtype rather than a general biomarker of PC. For instance, patients with germline mutations (e.g., *CDKN2A* and *BRCA2*) or IPMN may have distinct molecular mechanisms for carcinogenesis. A combination of different cfDNA aberrations (e.g., mutations and chromosomal aberrations) may result in a panel of markers with higher sensitivity. Thus, while this case-control study is not able to conclusively prove a role for CNV testing in early PC detection, it highlights its potential and should be regarded as the starting point for further research in these specific surveillance cohorts.

In conclusion, shallow sequencing using the robust NIPT pipeline is feasible for PJ cfDNA analysis. We identified 8q24 and the 2q-5p combination as hotspots, which seem specific for PC. Future studies with larger sample sizes are required, including parallel testing of tissue and cfDNA analysis of PJ samples. Additionally, to evaluate the role of PJ cfDNA analysis in pancreas surveillance, the analysis of consecutive samples to assess the evolution of cfDNA changes over time is essential.

## 4. Materials and Methods

### 4.1. Study Design and Patient Inclusion

This study included data and biomaterial that were prospectively collected at the Erasmus Medical Center in Rotterdam as part of three clinical studies: (1) KRASPanc study (MEC-2018-038), concerning patients with (suspected) sporadic PDAC undergoing diagnostic endoscopic ultrasound (EUS) or fiducial placement for stereotactic radiotherapy; (2) PACYFIC study (MEC-2014-021), involving individuals undergoing surveillance for suspected neoplastic pancreatic cysts; (3) FPC study, including individuals with a hereditary increased risk of pancreatic cancer undergoing surveillance. See Appendix A for the in- and exclusion criteria per study. The institutional ethical review board approved these studies, and they were carried out according to the ethical principles for medical research involving human subjects from the World Medical Association Declaration of Helsinki. Participants provided written informed consent before enrolment. All authors had access to the study data and reviewed and approved the final manuscript.

The current study consisted of two phases. The first (“feasibility”) phase was executed to evaluate the feasibility of chromosomal aberration detection in cfDNA isolated from PJ by comparing the results with matched plasma and/or tissue samples (analyzed by chromosomal microarray) from three patients with pathology-proven PC and one individual with pathology-proven intraductal papillary mucinous neoplasm (IPMN) harboring high-grade dysplasia (HGD).

The second (“experimental”) phase included PJ samples from individuals from the feasibility phase, additional cases with sporadic PC who underwent PJ collection as part of the KRASPanc and controls undergoing pancreas surveillance without morphological changes on imaging as part of the FPC study (also known as the CAPS study). The sample inclusion was based on the availability of PJ and, for the controls, the absence of morphological aberrations (no major Rosemont criteria and, at maximum, 1 minor Rosemont criterion, side or main duct < 5 mm and no absolute or relative indications for surgery) or clinical symptoms. No formal sample size analysis was performed due to the explorative nature of the study.

### 4.2. Biomaterial Collection

The PJ collection was performed with a linear echoendoscope (Pentax Medical, Tokyo, Japan) by experienced endo-sonographers. The secretion of PJ was stimulated by intravenous injection of human secretin (16 µg/patient, ChiRhoClin, Burtonsville, MD, USA) after positioning the tip of the scope close to the ampullary orifice. Suction through the endoscopic channel was applied (without occluding the ampullary orifice) for eight minutes directly after secretin infusion [11]. After collection, the PJ was aliquoted and snap-frozen. The samples were stored at −80 °C until use. On the same day as the PJ samples, the plasma samples were collected by venipuncture in Cellsave tubes, centrifuged, aliquoted and stored at −80 °C until use. The tissue was snap-frozen within hours after pancreatic resection.

### 4.3. cfDNA Analysis

To assess the presence of chromosomal aberrations in the plasma and PJ, a diagnostic routine NIPT pipeline was used. As PJ samples typically harbor higher cfDNA concentrations than blood plasma, the PJ samples were diluted with ice-cold PBS, ensuring that an input of ca. 1 ng cfDNA was used for the automated pipeline. When necessary, the input was diluted so that the pipeline could accept the sample for the run. After the samples were cold centrifuged at 1600× *g* and 4 °C for 10 min, an automated NGS workflow was performed using the VeriSeq NIPT microlab Star robot (Hamilton, Gräfelfing, Germany). In short, the plate was sealed and recentrifuged at 5600× *g* and 20 °C for 10 min. The DNA from 900 µL of supernatant was extracted, and the sequence library was created using the VeriSeq NIPT solution (Illumina, Cambridge, UK). Subsequently, a unique synthetic DNA “barcode” (index) was attached to each sample, and the library product was quantified using a fluorescent dye and comparison of the results with a DNA standard curve. At last, shallow whole-genome sequencing was performed on a NextSeq500 sequencer yielding 2 × 36 paired-end reads in a 48-plex reaction. The SeqFF model was used as a surrogate marker to assess the percentage of short fragmented cfDNA that were likely of tumor origin in the DNA pool (herein called the percentage tumor cfDNA) [63].

### 4.4. Microarray Analysis

To prepare the tumor tissue samples for CNV analysis, the tissue was washed twice with PBS and subsequently treated with 550 µL lysis buffer and 20 µL protease overnight at 37–55 °C. A total of 3 µL RNase A was added and incubated for 15–60 min at 37 °C. Subsequently, a Chemagic MSM1 isolation robot (PerkinElmer Chemagen Technology, Baesweiler, Germany) was used to isolate the DNA according to the manufacturer’s recommendations. The genotyping was performed on an SNP array (Illumina GSAMD-24v3 chip) according to the manufacturer’s recommendations (Illumina, Cambridge, UK).

Five PJ samples (1 case, 4 controls) were excluded from further analysis due to the fact of technical failure of the Veriseq NIPT microlab Star robot, even upon repeated measurements. These samples were not included in the described analyses. In addition, the samples with <5 million reads were included in the fragment length analysis, yet excluded from further CNV (and sensitivity) analysis, as this was considered indicative of an insufficient number of reads for calling chromosomal aberrations. The fragment size (insert size) distribution within the samples was evaluated (and compared) to that in blood plasma. All sequencing results were visualized on Wisecondor graphs, and the present gains and losses were assessed by an experienced reader (MIS) [64]. The findings were divided into four categories according to the type of chromosomal aberrations: (1) no significant CNVs; (2) samples with an 8q gain and/or 10p loss; (3) samples with a 2q and/or 5p loss; (4) samples with chaotic aberrations spread over multiple chromosomes without a clear relationship to each other. The genome reference database used in this study was GRCh37.

### 4.5. Statistical Analysis

The descriptive data were expressed as medians with interquartile range (IQR; continuous data) or percentages (categorical data). Statistical significance was assessed using the Mann–Whitney U and χ^2^ tests, respectively. The sensitivity, specificity and accuracy were calculated by receiver operating curve (ROC) analysis, and the confidence interval (CI) was represented as an “exact” Clopper–Pearson confidence interval. A two-sided *p*-value < 0.05 was considered statistically significant. The analyses were performed in SPSS (Statistical Package for the Social Sciences, version 27, SPSS Inc., Chicago, IL, USA); the figures were created using GraphPad (GraphPad Prism version 9, GraphPad Software, La Jolla, CA, USA).

## Figures and Tables

**Figure 1 ijms-24-05097-f001:**
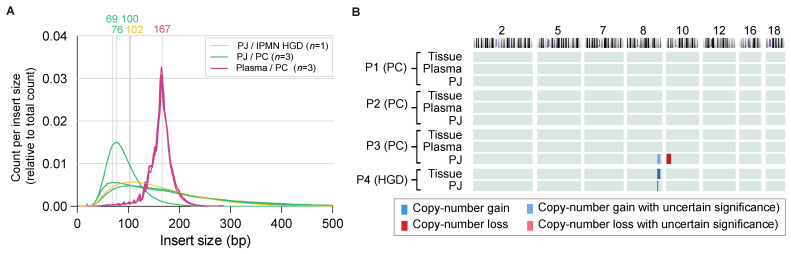
Results of pilot samples. (**A**) The fragment lengths’ distributions (insert size) in pilot PJ and plasma samples. No clear difference was seen between the intraductal papillary mucinous neoplasm (IPMN; yellow) with high-grade dysplasia (HGD) and pancreatic cancer (PC; green). PJ samples showed a mode of 69-102 base pairs (bp), whereas the fragment peak for the plasma samples (all PC) was 167 bp (pink). (**B**) Chromosomal aberrations in pilot samples. No significant aberrations were seen in plasma samples, whereas in pancreatic juice (PJ) an 8q gain with (P3) or without (P4) 10p loss was seen. For P4, this aberration was also found in tissue.

**Figure 2 ijms-24-05097-f002:**
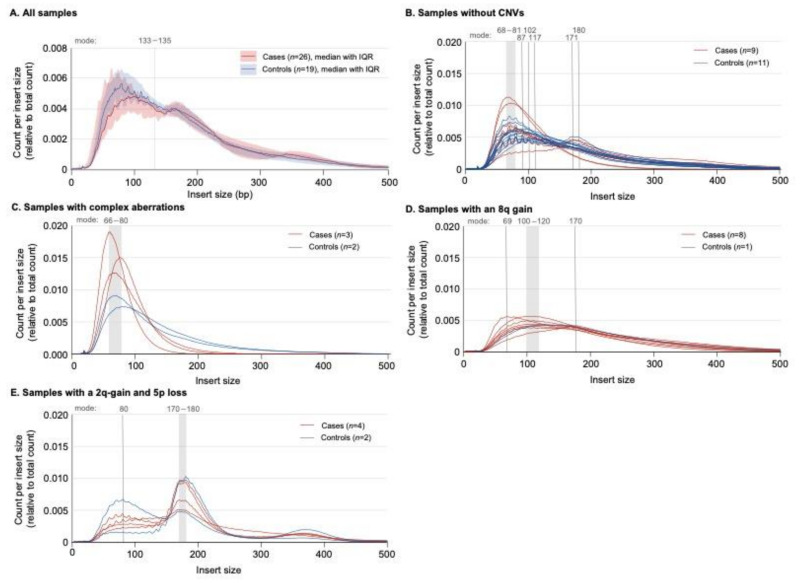
The fragment lengths’ (insert size) distributions in pancreatic juice (PJ) samples. (**A**) The cases (red) in comparison with the controls (blue); fragments with a length of 133–135 base pairs (bp) were more prevalent in the controls than cases (*p* = 0.05; Mann–Whitney U). IQR = interquartile range. (**B**) Samples without a copy number variation (CNV) on the Wisecondor image. The peak prevalence varied between 65 and 117, 171 and 180 bp. Ten out of twenty samples had a sawtooth pattern (50%), which all showed a showed a second peak at 170–180 bp. (**C**) Samples with complex (chaotic) aberrations on the Wisecondor image. Each sample harbored mostly short fragments (mode 65–80 bp). The presence of short fragments and the complex (chaotic) aberrations may be related. (**D**) The samples from individuals with an 8q gain showed a similar graph with a moderate (or no) sawtooth pattern. Six out of eight individuals had a mode between 100 and 120 bp. (**E**) Samples with a 2p gain and 5q loss: each line shows two peaks (at ±80 bp and 170–180 bp); the first peak has a clear sawtooth pattern.

**Figure 3 ijms-24-05097-f003:**
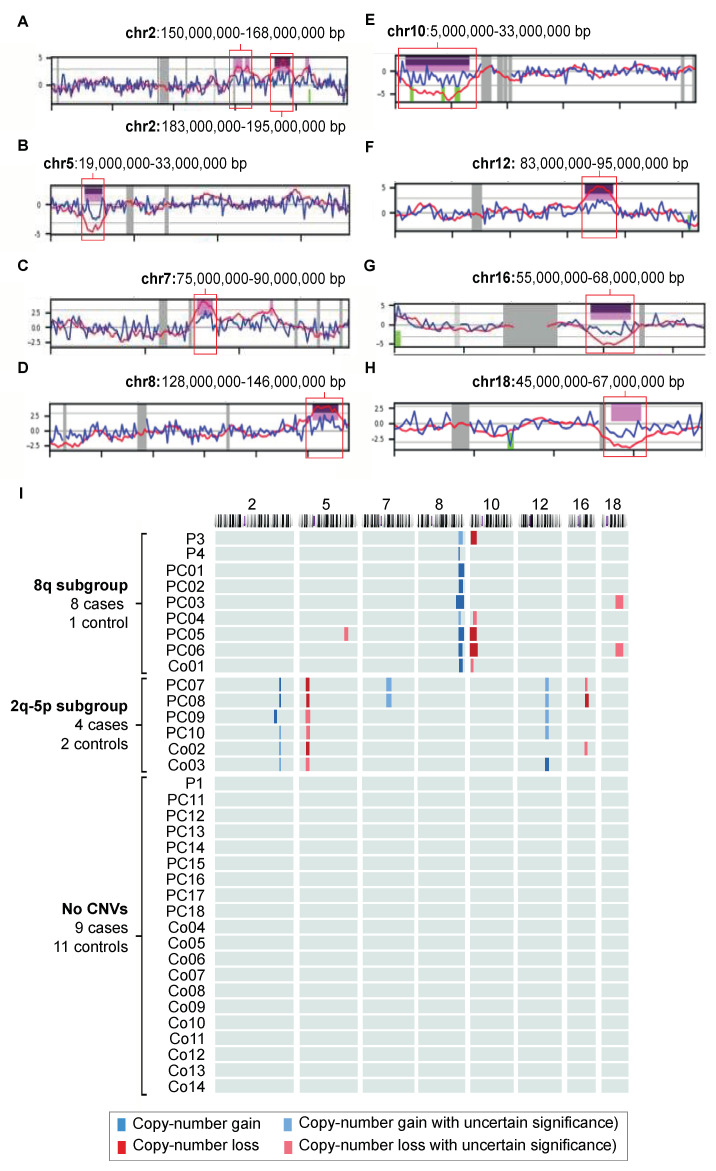
Copy-number gains (blue) and losses (red) per chromosome (**A**) and patient (**B**) of all pancreatic juice (PJ) samples included in the study. (**A**–**H**). Snapshots of Wisecondor images. Purple = aberration called by software; pink = aberration of uncertain significance. Locations on the chromosomes are given in base pairs (bp). (**I**) Nine individuals had an 8q gain (with or without a 10p loss and/or 18 loss), and six individuals had both a 2q gain and 5p loss (with or without a 7q gain, 12q gain and/or 16q loss). The aberrations in the subgroups did not overlap. P = pilot; PC = PC case; Co = control. Chromosomes without a significant aberration and individuals with a “chaotic profile” (P2, PC19, PC20, Co15 and Co16) are not shown.

**Table 1 ijms-24-05097-t001:** Patient characteristics.

	Cases (n = 26)	Controls (n = 19)	*p*-Value
Age, median (IQR)	69 (9)	60 (9)	<0.001
Sex, n male (%)	18 (69)	7 (37)	0.04
BMI, median in kg/m^2^ (IQR)	23 (4)	24 (5)	0.23
Diabetes mellitus, n present (%)	11 (42)	0 (0)	<0.001
Hereditary predisposition, n (%)	0 (0)	19 (100)	<0.001
Member of FPC family		7 (16)	
CDKN2A germline mutation		8 (18)	
BRCA1/2 germline mutation		3 (7)	
PALB2 germline mutation		1 (2)	
History of malignancy, n (%)	3 (12)	8 (42)	0.02
Breast cancer	0 (0)	1 (5)	
Melanoma	0 (0)	7 (37)	
Other	3 (12)	0 (0)	
Any symptom, n (%)	21 (81)	0 (0)	<0.001
Jaundice	8 (31)		
Epigastric pain	15 (58)		
Weight loss	11 (42)		
CA19.9 >37 kU/L, n (%)	19 (73)	NA	NA
Treatment-naive, n (%)	23 (88)	NA	NA
Resectability of PC, n (%)		NA	NA
Resectable	8 (31)		
Borderline resectable	2 (8)		
Locally advanced PC	16 (62)		
Location mass, n (%)		NA	NA
Uncinate/head	15 (58)		
Neck/corpus	8 (31)		
Tail	3 (12)		
*CBD stent* in situ, n (%)	7 (27)	NA	NA
Surgery, n (%)	11 (42)	NA	NA
Pancreaticoduodenectomy	8 (31)		
Distal pancreatectomy	3 (12)		

**Table 2 ijms-24-05097-t002:** Diagnostic performance of the chromosomal profiles. AUC = area under the curve; CI = confidence interval.

	Present/Total	AUC (95% CI)	Sensitivity in %	Specificity in %	Accuracy in %
8q-10p profile	9/40	0.64 (0.46–0.81)	33 (16–55)	94 (70–100)	57 (41–73)
2q-5p profile	6/40	0.52 (0.34–0.70)	17 (5–38)	87 (62–98)	45 (29–62)
8q-10p or 2q-5p	15/40	0.66 (0.48–-0.83)	50 (29–71)	81 (54–96)	63 (46–77)

## Data Availability

Data is unavailable due to privacy or ethical restrictions.

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
