# Peer review of "An 8q24 Gain in Pancreatic Juice Is a Candidate Biomarker for the Detection of Pancreatic Cancer"

_ijms, 2023, doi:10.3390/ijms24065097_

Round 1

Reviewer 1 Report

-The results indeed demonstrate that detection of CNV's in PJ is feasible and has a high specificity for malignancy. To my knowledge, this is the first study that demonstrates this in PJ obtained from the duodenum upon secretin stimulation. Therefore tis study ratifies a trend towards non invasive screening modalities in high risk individuals for pancreatic cancer and provides a solid rationale for expanding future research in this field. 

- However due to the relative low sensitivity the identified CNV's may not be sufficient for screening if not combined with other biomakers/ screening modalities. Could the authors remark on this?

-The identified CNV's provide some novel insights in the molecular pathogenesis of oncogenic transformation en sequence of mutations, in particular MYC signaling

Minor questions:

- Fig 2A: based on the graph, controls seem to have an insert size 133-135 more often than cases. This does not match with the text where it is stated that controls less often had a fragment length of 133-135.

- The order of figure 3 (A-I) does not seem logical, as it does not match with the text. 

Author Response

Dear Reviewer, 

Thank you for your work on revising the manuscript. We are pleased to read that you accepted the manuscript ‘with minor revisions’. 

  1. In response to: 'However due to the relative low sensitivity the identified CNV's may not be sufficient for screening if not combined with other biomakers/ screening modalities. Could the authors remark on this?' Thank you for this remark. I totally agree. A sensitivity of 33% is not sufficient for usage as a single test, yet could be valuable in conjunction with other features, such as absolute and relative indications for surgery (as described in European evidence-based guidelines on pancreatic cystic neoplasms; doi: 10.1136/gutjnl-2018-316027) and genetic mutations. 
  2. In response to: 'Fig 2A: based on the graph, controls seem to have an insert size 133-135 more often than cases. This does not match with the text where it is stated that controls less often had a fragment length of 133-135.' Thank you for highlighting this. I made the following change in the text: 'Controls less often...' was amended and is now 'Controls more often..'
  3. In response to: 'The order of figure 3 (A-I) does not seem logical, as it does not match with the text.' I amended the sequence of the Figures and the flow of the text is better now. The first part of 2.4 8q24 subgroup is now: 'Nine individuals (23%; 8 cases [33%], 1 control [6%], p=0.04) had at least an 8q24 gain (chr8:128,000,000-146,000,000; Figure 3A,B). DNA insert size mode was 100-120bp in 7/9 samples (69bp and 170 bp in the other two) and all graphs showed a smooth shape (Figure 2D). Other CNVs present in this subgroup were 10p loss (n=5; chr10: 8,000,000-25,000,000; 10p11-1p15; Figure 3A,C), 18q loss (n=2; chr18: 45,000,000-67,000,000; 18q21-18q22; Figure 3A,D) and a single 5q gain (chr5: 143,000,000-155,000,000; 5q31-5q33)...'. The same change was made in the text in the 2q-5p paragraph refering to Figure 3E to 3I.  

It seems that you have not received the Supplemental Figures. As not Many changes were made to the main manuscript, I have decided to upload the supplemental Figure and table file for review (rather than the main manuscript, as only 1 file can be uploaded). I can also send the amended new manuscript, if needed.

Again, thank you so much for accepting this manuscript with minor revisions. Looking forward to your response.

Your sincerely, 

Iris Levink

Reviewer 2 Report

CNV analysis of cfDNA in pancreatic juice, which is attempted in this paper, provides important information for the diagnosis of pancreatic cancer. I believe that this paper deserves to be accepted. However, some additions are necessary to deepen the reader's understanding. Regions where CNV was markedly observed, including 8q24, and other areas where significant changes were observed, should be shown genomic maps which display the genes found in the region. In such a figure, it will be necessary to show which gene is related to cancer and which gene are not by color coding.

Author Response

Dear Reviewer, 

Thank you for your work on revising the manuscript. We are pleased to read that you accepted the manuscript ‘with minor revisions’. 

In response to: 'Regions where CNV was markedly observed, including 8q24, and other areas where significant changes were observed, should be shown genomic maps which display the genes found in the region. In such a figure, it will be necessary to show which gene is related to cancer and which gene are not by color coding.'

This has been highlighted in our supplemental Figures. Could it be possible that you have not received these? Please find them attached to this response. 

Your sincerely, 

Iris Levink